# Air quality trends and regimes in South Korea inferred from 2015−2023 surface and satellite observations

Yujin J. Oak[1], Daniel J. Jacob[1,2], Drew C. Pendergrass[1], Ruijun Dang[1], Nadia K. Colombi[2], Heesung Chong[3], Seoyoung Lee[4,5], Su Keun Kuk[6], Jhoon Kim[7]

[1] School of Engineering and Applied Sciences, Harvard University, Cambridge, MA, USA
[2] Department of Earth and Planetary Sciences, Harvard University, Cambridge, MA, USA
[3] Harvard-Smithsonian Center for Astrophysics, Cambridge, MA, USA
[4] Goddard Earth Sciences Technology and Research (GESTAR) II, University of Maryland, Baltimore County, Baltimore, MD, USA
[5] Climate and Radiation Laboratory, NASA Goddard Space Flight Center, Greenbelt, MD, USA
[6] Samsung Advanced Institute of Technology, Samsung Electronics Co., Ltd., Suwon, South Korea
[7] Department of Atmospheric Sciences, Yonsei University, Seoul, South Korea

*Correspondence to*: Yujin J. Oak (yjoak@g.harvard.edu)

**Abstract.**

We analyze 2015−2023 trends in air quality in South Korea using surface (AirKorea network) and satellite measurements, including the new GEMS geostationary instrument. Primary air pollutants (CO, $SO_2$, $NO_2$) have decreased steadily at rates consistent with the national CAPSS emissions inventory. Volatile organic compounds (VOCs) show no significant trend. GEMS glyoxal (CHOCHO) identifies large industrial sources of VOCs while formaldehyde (HCHO) points to additional biogenic sources. Surface ozone ($O_3$) peaks in May−June and the maximum 8-hour daily average (MDA8) exceeds the 60 ppbv standard everywhere. The AirKorea average May−June 90$^{th}$ percentile MDA8 $O_3$ increased at 0.8 ppbv $a^{-1}$, which has been attributed to VOC-sensitive conditions. Satellite HCHO/$NO_2$ ratios indicate that the $O_3$ production regime over Korea is shifting from VOC- to $NO_x$-sensitive conditions as $NO_x$ emissions decrease. The $O_3$ increase at AirKorea sites is because most of these sites are in the Seoul Metropolitan Area where vestiges of VOC-sensitive conditions persist; we find no such $O_3$ increases over the rest of Korea where conditions are $NO_x$-sensitive or in the transition regime. Fine particulate matter ($PM_{2.5}$) has been decreasing at 5% $a^{-1}$ in both AirKorea and satellite observations but the nitrate ($NO_3^-$) component has not been decreasing. Satellite $NH_3$/$NO_2$ ratios show that $PM_{2.5}$ $NO_3^-$ formation was $NH_3$-sensitive before 2019 but is now becoming $NO_x$-sensitive as $NO_x$ emissions decrease. Our results indicate that further $NO_x$ emission decreases in Korea will reap benefits for both $O_3$ and $PM_{2.5}$ $NO_3^-$ as their production is now dominantly $NO_x$-sensitive.

## 1. Introduction

South Korea experienced rapid development over the past 30 years with an annual average GDP growth rate of 5% (S. Song and G. Lee, 2020). This has resulted in high emissions of carbon monoxide (CO), sulfur dioxide ($SO_2$), nitrogen oxides ($NO_x \equiv NO + NO_2$), nonmethane volatile organic compounds (NMVOCs), and primary fine particulate matter ($PM_{2.5}$, smaller than 2.5 µm diameter) (Y. Kim and G. Lee, 2018). Subsequent atmospheric chemistry produces surface ozone ($O_3$) and additional $PM_{2.5}$, which are the main pollutants of concern for air quality. 30,000 premature deaths per year are presently attributed to air pollution in South Korea (hereafter referred to as Korea) (Oak et al., 2023; J. Choi et al., 2024). National air quality standards were tightened in 2018 for $O_3$ (60 ppbv maximum 8-hour daily average or MDA8) and for $PM_{2.5}$ (15 µg $m^{-3}$ annual, 35 µg $m^{-3}$ 24-hour). None of the sites in the AirKorea governmental surface network meet the $O_3$ standard as of 2022, and only 4% meet the 24-hour $PM_{2.5}$ standard, despite governmental efforts to decrease emissions.

The need to decrease emissions responsible for air pollution has been recognized since the 1980s, prompting early control policies to regulate solid fuel use and outdoor combustion, and promote clean fuels. This effectively reduced $SO_2$, CO, and directly emitted (primary) PM (Y. Kim and G. Lee, 2018). More recent efforts by the Korean Ministry of Environment (MOE) have targeted $NO_x$ emissions. However, $O_3$ pollution has been getting worse at a rate of 1.0–1.5 ppbv $a^{-1}$ over 2000–2021 (S. W. Kim et al., 2023). $PM_{2.5}$ has decreased though unevenly (J. Jeong et al., 2022; H. M. Lee et al., 2024; Pendergrass et al., 2022; 2024), with an increasing contribution from secondary components produced chemically in the atmosphere including secondary organic aerosol (SOA) and particulate nitrate ($NO_3^-$) (H. M. Lee et al., 2024).

Synoptic meteorology and transport from China also contribute to seasonal and long-term variations of pollutants over Korea. Photochemical $O_3$ production is largest during the summer months, but $O_3$ peaks in May–June due to the summer monsoon in July–August (H. M. Lee and R. Park, 2022). Wildfires, stratospheric intrusions, and transport from China also contribute to high $O_3$ levels during May–June (H. M. Lee and R. Park, 2022). $PM_{2.5}$ is highest during the colder months (October–March), due to increased energy consumption and stagnant conditions over the Korean peninsula (J. Jeong et al., 2024), but here again transport from China also makes an important contribution (D. Park et al., 2021). $PM_{2.5}$ pollution in China has decreased considerably over the past decade in response to emission controls (Zhai et al., 2019) and this has

decreased its influence on Korea (Bae et al., 2021). On the other hand, $O_3$ pollution in China has gotten worse (K. Li et al., 2021).

Formation of $O_3$ and secondary $PM_{2.5}$ depends on complex chemistry involving $NO_x$ and NMVOCs that would respond nonlinearly to emission controls. $PM_{2.5}$ $NO_3^-$ formation further depends on ammonia ($NH_3$) emissions, which are mainly from agriculture and have not been decreasing. The dependences of $O_3$ and $PM_{2.5}$ concentrations on precursor emissions define chemical regimes that are important to understand for emission control strategies. They can be studied with 3-D chemical transport models (CTMs) that couple emissions, chemistry, and transport (R. Park et al., 2021). The formaldehyde (HCHO) to $NO_2$ ratio measured from satellite can diagnose $O_3$ sensitivity to VOCs versus $NO_x$ emissions (Duncan et al., 2010; Martin et al., 2004), and the $NH_3$ to $NO_2$ ratio can diagnose $NO_3^-$ sensitivity to $NH_3$ versus $NO_x$ emissions (Dang et al., 2023; 2024).

Satellites offer a growing resource for monitoring air pollutants, trends, and regimes over Korea. Low-Earth orbit (LEO) instruments observe at specific times of day. Important instruments include MOPITT (Edwards et al., 2004) and TROPOMI (Veefkind et al., 2012) for CO, OMI (Levelt et al., 2006) and TROPOMI for $SO_2$, $NO_2$, HCHO, and glyoxal (CHOCHO), and IASI (Van Damme et al., 2014) for $NH_3$. Geostationary instruments over East Asia including GOCI and GOCI-II provide hourly observations of aerosol optical depth (AOD) (M. Choi et al., 2018; S. Lee et al., 2023). The Geostationary Environment Monitoring Spectrometer (GEMS), launched in February 2020, provides the first hourly observations of gases by solar backscatter including $SO_2$, $NO_2$, HCHO, and CHOCHO (J. Kim et al., 2020).

Here we analyze recent 2015−2023 trends in air quality in Korea by exploiting both satellite and surface observations. We interpret the trends in terms of the major drivers and evaluate consistency with annual bottom-up emission estimates from the Clean Air Policy Support System (CAPSS) of the MOE (S. Choi et al., 2022). We start from 2015 when $PM_{2.5}$ observations from the AirKorea network became available, with subsequent milestones including the May−June 2016 Korea-United States Air Quality (KORUS-AQ) field campaign (Crawford et al., 2021) and satellite observations from TROPOMI (starting in May 2018) and GEMS (starting in November 2020). We use $HCHO/NO_2$ and $NH_3/NO_2$ indicators from the satellite data to diagnose $O_3$ and $PM_{2.5}$ chemical regimes and their trends.

## 2. Air quality observing system for South Korea

We make use of air quality observations in Korea from surface sites, aircraft, and satellites. The National Institute of Environmental Research (NIER) operates the AirKorea surface network of 642 monitoring sites as of 2023 (https://www.airkorea.or.kr/eng, last access: 12 August 2024), providing hourly data on CO, $SO_2$, $NO_2$, $O_3$, $PM_{10}$ (smaller than 10 µm diameter), and $PM_{2.5}$ concentrations. Monthly VOCs data (56 species) are available at a few urban sites. The KORUS-AQ field campaign in May−June 2016 included a detailed chemical payload onboard the DC-8 aircraft with extensive vertical profiling over the Seoul Metropolitan Area (SMA) at different times of day (Crawford et al., 2021). This was used by Yang et al. (2023) to infer diurnal profiles of $NO_2$ vertical column densities (VCDs) over the SMA and we will do the same here for HCHO and CHOCHO.

Satellite observations for air quality over Korea used in this work are compiled in Table 1 and are applied to analyze annual, diurnal, and spatial variations of pollutants. We filter out cloudy scenes using a cloud fraction threshold of 0.3 and apply additional quality filtering as recommended by the retrieval teams. GOCI and GOCI-II AOD retrievals are for 550 nm wavelength. CO is retrieved in both the shortwave and thermal infrared (SWIR and TIR). $NH_3$ is retrieved in the TIR. All other gases are retrieved in the ultraviolet-visible (UV-VIS). Tropospheric $O_3$ can also be retrieved in the UV but the measurements are difficult because of air scattering and the stratospheric column overhead, and different products are inconsistent over Korea (Gaudel et al., 2018). We do not use them here.

**Table 1. Satellite observations used in this work.**

| Instrument | Launch | Species[a] | Spatial resolution[b] | Version | Reference |
|---|---|---|---|---|---|
| *Low Earth orbit* | | | | | |
| MOPITT | 1999 | CO | $22 \times 22$ km$^2$ | V9 | Deeter et al. (2022) |
| OMI | 2004 | $SO_2$, $NO_2$, HCHO, CHOCHO | $13 \times 24$ km$^2$ | V3[c] | González Abad et al. (2015); Krotkov et al. (2017); C. Li et al. (2020); Kwon et al. (2024) |

| | | | | | |
|---|---|---|---|---|---|
| TROPOMI | 2017 | CO , $NO_2$, HCHO | 5.5 × 3.5 $km^2$ | V2.4.0 | De Smedt et al. (2018); Landgraf et al. (2016); van Geffen et al. (2022) |
| IASI | 2006 | $NH_3$ | 12 × 12 $km^2$ | V4 | Clarisse et al. (2023) |
| *Geostationary orbit* | | | | | |
| GEMS | 2020 | $SO_2$, $NO_2$[d], HCHO, CHOCHO | 3.5 × 7.7 $km^2$ | V2.0.0 | Ha et al. (2024); G. T. Lee et al. (2024); NIER (2020) ; Oak et al. (2024) |
| GOCI | 2011 | AOD | 2 × 2 $km^2$ | YAER[e] V2 | M. Choi et al. (2018) |
| GOCI-II | 2020 | AOD[f] | 2.5 × 2.5 $km^2$ | YAER | S. Lee et al. (2023) |

[a] Total atmospheric columns except for $NO_2$ (tropospheric column).
[b] Native pixel resolution of retrieval.
[c] Provided at 1° × 1° by Kwon et al. (2024).
[d] Bias-corrected by Oak et al. (2024).
[e] Yonsei Aerosol Retrieval.
[f] Observations within the range of GOCI AOD (−0.05 to 3.6) are used to account for the systematic low bias in GOCI-II compared to GOCI (S. Lee et al., 2023; Pendergrass et al., 2024).

## 3. Air quality distributions and trends in South Korea

Here we analyze spatial distributions and temporal trends of individual air pollutants using surface and satellite observations, and compare the trends to the annual bottom-up estimates of anthropogenic emissions from CAPSS, reported with a two-year lag (https://www.air.go.kr/eng/main.do, last access: 12 August 2024). CAPSS includes city/county/district (Korean; si/gun/gu) level emissions for source categories including fuel combustion, manufacturing, solvent use, mobile sources, agriculture, and anthropogenic biomass burning (biofuel, agriculture).

Figure 1 shows major anthropogenic source regions in Korea. There are seven major cities with populations larger than one million. The SMA (37–37.8° N, 126.4–127.5° E) is the largest urban area which includes Seoul, Incheon, and surrounding suburbs, with concentrated

electronics and chemical industry. The southeast region including Busan and Ulsan is the second largest urban area and has petrochemical facilities, oil refineries, and steel/ship/automobile manufacturing industries.

### 3.1. Carbon monoxide (CO)

CO levels in Korea have consistently remained below the national air quality standards (9 ppmv 8-hour, 25 ppmv 1-hour) since the late 1990s (NIER, 2023). CO is nevertheless a useful tracer of pollution and plays an important role driving ozone formation in Korea (Gaubert et al., 2020; H. Kim et al., 2022). Anthropogenic CO emissions in CAPSS are 45% from transportation (passenger vehicles, heavy-duty vehicles, ships) and 32% from biomass burning (agricultural waste incineration, biofuels). Figures 2a−c compare 2021 CAPSS CO emissions with 2023 average surface CO and TROPOMI VCDs. Concentrations are highest in urban and industrial areas. Low VCDs along the east coast are due to topography. The effect of topography on VCDs is more apparent for CO than for other species because of the longer lifetime of CO and hence higher background (lower variability).

Figure 2d shows annual trends, demonstrating consistency between CAPSS and atmospheric observations. CAPSS emissions and AirKorea surface concentrations decrease at similar rates of $-2.3 \pm 1.7$ and $-2.6 \pm 0.7\%\ a^{-1}$. MOPITT decreases at a rate of $-0.9 \pm 0.5\%\ a^{-1}$, slower than surface concentrations because of the background contribution to the VCD. Chong et al. (2023) previously found a MOPITT CO decrease of $-0.6 \pm 0.1\%\ a^{-1}$ during 2005−2018. It is estimated that Chinese emissions contributed 21−25% to the downward trend between 2016 and 2022 (J. Park et al., 2024; E. Kim et al., 2024). The 2019 spike found in both surface CO and VCDs is due to stagnant conditions (J. Cho et al., 2022). This also affected other pollutants as will be shown below.

### 3.2. Sulfur dioxide ($SO_2$)

$SO_2$ levels in Korea have consistently remained below the national air quality standards (20 ppbv annual, 50 ppbv 24-hour) over the past two decades due to large reductions of emissions from power plants and the petrochemical industry (NIER, 2023). There is continuing motivation for $SO_2$ emission controls to decrease $PM_{2.5}$ sulfate ($SO_4^{2-}$). Figures 3a−c compare 2021 CAPSS $SO_2$ emissions with 2023 average surface $SO_2$ and GEMS VCDs for all available observations.

GEMS displays enhancements in the SMA, mid-south coast (power plants, petrochemical/steel industry) and northeastern regions (cement/concrete/pulp industry), consistent with previously (2011–2016) identified OMI $SO_2$ hotspots (Chong et al., 2020).

Figure 3d shows good agreement between the CAPSS-reported emission trends and atmospheric observations. CAPSS-reported emissions have decreased at a rate of –9.9 ± 3.3% $a^{-1}$, while surface $SO_2$ concentrations and OMI VCDs have decreased at similar rates of –6% $a^{-1}$ since 2015. Past trends (1999–2016) in Seoul showed that local emissions were the main drivers of the long-term decrease in surface $SO_2$ (J. Seo et al., 2018). J. Park et al. (2024) found that national mean surface $SO_2$ decreased by 41% from 2016 to 2022, owing to reductions in domestic (25%) and Chinese (16%) emissions.

### 3.3. Nitrogen dioxide ($NO_2$)

$NO_2$ levels exceeded the national standards (30 ppbv annual, 60 ppbv 24-hour) at 28% of the AirKorea sites in 2015 but fewer than 1% in 2022 (NIER, 2023). $NO_x$ emissions in Korea are dominated by the transportation sector, accounting for 64% of the CAPSS inventory. Control of $NO_x$ emissions is more recent than for CO and $SO_2$ and has been motivated not only by the $NO_2$ standards but also to reduce $PM_{2.5}$ $NO_3^-$. CAPSS $NO_x$ emissions declined by 23% from 2015 to 2021 in response to policies including stronger regulation on heavy-duty diesel engines in 2016 (S. Song and G. Lee, 2020) and seasonal PM management plans implemented in 2019 (Bae et al., 2022; J. Jeong et al., 2024).

Figures 4a–c compare 2021 CAPSS $NO_x$ emissions with 2023 average surface $NO_2$ and GEMS tropospheric VCDs. Here we use a GEMS product calibrated to TROPOMI to remove artifacts (Oak et al., 2024). Surface concentrations and VCDs display similar spatial distributions, with highest values in the SMA and other urban areas in the southeast. Figure 4d shows that surface $NO_2$ and OMI tropospheric VCDs have decreased over the 2015−2023 period by 32% and 36%, respectively. The trend in CAPSS-reported emissions (−4.8 ± 2.7% $a^{-1}$) is consistent with surface observations (−4.4 ± 0.8% $a^{-1}$) and OMI VCDs (−4.6 ± 0.8% $a^{-1}$) during 2015−2023. Meteorology-corrected trends in tropospheric VCDs observed by ground-based remote sensing instruments at urban sites decreased at similar rates (−5.0 to −5.4% $a^{-1}$) during 2015−2020 (Y. Choi et al., 2023). Long-term (2005−2019) records show that significant decreases in surface and OMI $NO_2$ began in 2015 (S. Seo et al., 2021). CAPSS shows in increase

from 2015 to 2016, which is due to updates in emission factors (S. Choi et al., 2020). E. Kim et al. (2024) found that only 2% of the observed 23% decrease in surface $NO_2$ during 2016−2021 over Korea was attributable to the Chinese contribution.

Geostationary satellite observations provide additional information on diurnal variation. Figure 4e shows the 2021−2023 seasonal mean hourly variations of surface $NO_2$ and GEMS VCDs over the SMA. Both surface and column $NO_2$ are higher by a factor of two during the cold season, which can be explained by the longer $NO_x$ lifetime (Shah et al., 2020). Surface $NO_2$ concentrations peak at 8−9 local time (LT) when daytime emissions accumulate in a shallow mixed layer, then decrease by dilution over the rest of the morning as the mixed layer grows from solar heating, returning to a secondary maximum in the evening when the mixed layer collapses (Moutinho et al., 2020). In contrast, VCDs increase steadily in the morning as they are not affected by mixed layer growth, reaching a steady state in the cold season as daytime emissions become balanced by ventilation, and an afternoon decrease in the warm season due to the additional effect of the daytime photochemical sink (Yang et al., 2024).

### 3.4. Nonmethane volatile organic compounds (NMVOCs)

NMVOCs emissions include important contributions from both anthropogenic and biogenic sources. More than half of anthropogenic VOCs (AVOCs) emissions in CAPSS are from solvent use while transportation is responsible for less than 10%, although the latter may be a severe underestimate (S. Song et al., 2019; Y. Kim and G. Lee, 2018; Kwon et al., 2021). CAPSS also does not account for residential emissions of volatile chemical products (VCPs), which could be large in Korea as indicated by observations of elevated ethanol during KORUS-AQ (Beaudry et al., 2024; Travis et al., 2024). Annual total AVOCs emissions are estimated to be a factor of two larger than biogenic VOCs (BVOCs) on a national level (Jang et al., 2020). However BVOCs play an important role in $O_3$ and SOA formation during summer (H. K. Kim et al., 2018; Oak et al., 2022; H. M. Lee and R. Park, 2022), when its emissions are comparable to those of AVOCs (J. Choi et al., 2022).

Figures 5a–b compare 2021 total AVOCs emissions from CAPSS and BVOCs emissions calculated from MEGAN (Model of Emissions of Gases and Aerosols from Nature) (Guenther et al., 2012). The two have contrasting distributions, with AVOCs mostly urban and industrial. Shown in Figure 5c is the distribution of BTEX (≡ benzene + toluene + ethylbenzene + xylenes)

concentrations observed at AirKorea sites, with high values over urban areas consistent with CAPSS. Benzene is elevated on the west and southern coasts where it originates from the steel industry, oil refineries, and petrochemical facilities (Fried et al., 2020; C. Cho et al., 2021; Y. Seo et al., 2014). Toluene, xylenes, and ethylbenzene are abundant in the SMA (Y. Lee et al., 2023; S.-J. Kim et al., 2021; S. Song et al., 2019) due to emissions from traffic and solvent use (Simpson et al., 2020).

Figures 5d–e show spatial distributions of HCHO and CHOCHO VCDs from GEMS. These are common intermediates in the oxidation of NMVOCs, but CHOCHO is preferentially produced from aromatics (Kaiser et al., 2015; J. Li et al., 2016). Satellite observations are most sensitive to precursor NMVOCs with short lifetimes and prompt HCHO or CHOCHO yields including isoprene, alkenes, toluene, and xylenes (Palmer et al., 2003; Bates et al., 2021; Chan Miller et al., 2017). The GEMS CHOCHO and HCHO VCDs are elevated in major industrial regions, but CHOCHO shows hotspots for manufacturing industries while HCHO shows hotspots for petrochemical facilities. HCHO observations are also more distributed, reflecting the larger BVOCs contribution from isoprene.

Figure 5f shows the CHOCHO to HCHO ratio $R_{GF}$ = $VCD_{CHOCHO}/VCD_{HCHO}$, illustrating the contrast in their sources. $R_{GF}$ is generally higher under anthropogenic dominance (Chen et al., 2023). Values range from 0.02 in rural regions to more than 0.05 in the SMA and Busan. In the US, $R_{GF}$ values are below 0.03 even under polluted conditions (Chan Miller et al., 2017) and are down to 0.01 in rural regions with dominant biogenic sources (Kaiser et al., 2015). GEMS $R_{GF}$ values in Korea are higher everywhere, indicating a more important role for AVOCs emissions than in the US where these emissions have been strongly regulated for decades (Parrish et al., 2009; Warneke et al., 2012). Unlike for other pollutants and in contrast to the US, regulation of AVOCs emissions in Korea has been limited (S. Song and G. Lee, 2020; J. Kim et al., 2023). Figure 5g shows no significant trends in AVOCs emissions, surface BTEX, and satellite observations of CHOCHO and HCHO from OMI, TROPOMI and GEMS during 2015–2023.

Figure 6 compares diurnal variations of HCHO and CHOCHO VCDs in the SMA observed from GEMS and DC-8 aircraft profiles during KORUS-AQ (May−June 2016). Here we use airborne observations conducted below 8 km over the SMA. Mean loss frequencies of HCHO and CHOCHO against oxidation by OH and photolysis average 0.42 $h^{-1}$ and 0.61 $h^{-1}$, respectively at 11−15 local time in these aircraft profiles. Computation of VCDs and loss

frequencies from the KORUS-AQ data is described in the Supplement. We find that the GEMS columns are lower than the aircraft column and this has been previously reported as systematic low biases in satellite observations of CHOCHO and HCHO (Chan Miller et al., 2017; Zhu et al., 2016; Zhu et al., 2020). HCHO VCDs are more than twice higher during the warm season (April−September) than the cold season (October−March), consistent with a biogenic contribution to HCHO, while CHOCHO VCDs show no seasonal difference. GEMS and aircraft diurnal variations show HCHO and CHOCHO increases in the morning from photochemical production (G. T. Lee et al., 2024), flattening by midday. The aircraft data show a late afternoon rise in HCHO but that is not seen in the satellite data.

### 3.5. Ozone ($O_3$)

None of the AirKorea monitoring sites met the MDA8 standard of 60 ppbv for $O_3$ as of 2022 (NIER, 2023). $O_3$ peaks in May–June in Korea (Figure 7a) with contributions from domestic emissions, wildfires, stratospheric intrusions, and transport from China (H. M. Lee and R. Park, 2022). Several studies have reported on the $O_3$ increase in Korea over the past two decades, using different $O_3$ concentration metrics and time periods (J. Seo et al., 2018; Yeo and Kim, 2022; S. W. Kim et al., 2023). Our own analysis of the May–June 90$^{th}$ percentile MDA8 $O_3$ calculated for individual AirKorea sites and then averaged across all sites shows a rapid increase of 1.5 ± 0.4 ppbv $a^{-1}$ for 2005–2014, and a slower rate of 0.8 ± 0.9 ppbv $a^{-1}$ for 2015–2023 (Figure 7b).

Previous studies found that $O_3$ formation in major cities in Korea is in the VOC-sensitive regime, where decreasing $NO_x$ emissions causes $O_3$ to increase (S. Kim et al., 2018; S. W. Kim et al., 2023; Oak et al., 2019; Souri et al., 2020; H. J. Lee et al., 2021). However, as $NO_x$ emissions have decreased (Figure 4) whereas VOC emissions have not (Figure 5), $O_3$ formation may shift to a $NO_x$-sensitive regime. The HCHO to $NO_2$ column ratio ($R_{FN}$ = $VCD_{HCHO}/VCD_{NO2}$), an indicator for $O_3$ sensitivity to $NO_x$ versus VOCs (Duncan et al., 2010; Martin et al., 2004), increased steadily from 2015 to 2023 as seen from OMI, TROPOMI, and GEMS (Figure 7c). Based on the criteria from Duncan et al. (2010) the positive trend in $R_{FN}$ implies that Korea is now mostly in the $NO_x$-sensitive regime ($R_{FN}$ > 2). Figures 7d–e show May−June 2023 MDA8 $O_3$ and its sensitivity regimes inferred from GEMS $R_{FN}$. Most of the country is in a $NO_x$-sensitive regime while VOC-sensitive conditions are largely limited to the

central SMA. The broader SMA and urban southeastern Korea are in a transition regime where $O_3$ is sensitive to both $NO_x$ and VOCs emissions. These latter regions experience the most severe $O_3$ pollution as both $NO_x$ and VOCs contribute to $O_3$ formation.

Also shown in Figure 7b are May−June MDA8 $O_3$ trends for AirKorea sites in different sensitivity regimes based on the 2023 GEMS $R_{FN}$. The $O_3$ increase during 2015−2023 is only found in the VOC-sensitive areas (1.6 ± 0.8 ppbv $a^{-1}$). $O_3$ in $NO_x$-sensitive areas does not show any noticeable increase. Reports of $O_3$ increases in Korea based on data from the AirKorea sites may be biased by the AirKorea sites being concentrated in the SMA, which has been mostly VOC-sensitive. But this is now changing as $NO_x$ emissions decrease, and $O_3$ pollution in Korea is now poised to decrease everywhere in response to continued $NO_x$ emission controls. In the US, national average $O_3$ levels started to level off in the 1990s and declined significantly afterwards, shifting from VOC- to $NO_x$-sensitive regimes in response to $NO_x$ reduction (He et al., 2020). The 2023 US national average May−September 90$^{th}$ percentile MDA8 $O_3$ is now slightly above 60 ppbv (US EPA, 2024). An additional challenge for Korea to meet its air quality standard is the high background originating from East Asia, estimated to be 55 ppbv in May−June (Colombi et al., 2023).

### 3.6. Particulate matter (PM)

PM levels have steadily decreased in Korea over the 2015–2023 period with more than 95% of the AirKorea sites meeting the annual $PM_{10}$ standard (50 μg $m^{-3}$) since 2018. However, only 27% of sites met the $PM_{2.5}$ annual standard (15 μg $m^{-3}$) in 2022, and only 4% met the 24-hour standard (35 μg $m^{-3}$) (NIER, 2023). Figures 8a–c show that $PM_{10}$, $PM_{2.5}$, and GOCI AOD share similar spatial distributions. Annual trends in $PM_{10}$ (−4.0 ± 1.7% $a^{-1}$), $PM_{2.5}$ (−5.0 ± 1.6% $a^{-1}$), and AOD (−5.5 ± 2.7% $a^{-1}$) over Korea during 2015−2023 are consistent (Figure 8d). J. Park et al. (2024) found that 14% of the observed 33% decrease in $PM_{2.5}$ during 2016−2022 over Korea was attributable to the Chinese contribution.

Figure 8e shows seasonal mean hourly variations of surface $PM_{2.5}$ and GOCI AOD. Surface $PM_{2.5}$ peaks in winter to early spring, mostly attributable to sulfate-nitrate-ammonium aerosols (Zhai et al., 2021) and is minimum in summer during the monsoon period (H. M. Lee et al., 2024). Conversely, AOD peaks in spring and summer (March−August) due to dust events, chemical production of secondary aerosols, and hygroscopic growth at high relative humidity

(Zhai et al., 2021). $PM_{2.5}$ peaks at 9−11 LT local time and then decreases until late afternoon as the mixed layer grows and dilutes surface concentrations (Jordan et al., 2020). AOD rises in the morning and peaks in midday reflecting photochemical production (Lennartson et al., 2018; P. Kim et al., 2015).

2015−2021 $PM_{2.5}$ observations in Seoul shows that all major $PM_{2.5}$ components decreased except for $NO_3^-$, which accounts for 25% of total $PM_{2.5}$ during winter to early spring (H. M. Lee et al., 2024). Winter $NO_3^-$ formation depends non-linearly on $NO_x$ and $NH_3$ emissions, with dominant sensitivity to either precursor that can be diagnosed from the $NH_3/NO_2$ VCD ratio and the $NO_2$ VCD in satellite observations (Dang et al., 2023; 2024). Figures 9a−b compare 2021 CAPSS $NH_3$ emissions and 2023 average $NH_3$ VCDs observed by IASI. 76% of anthropogenic $NH_3$ emissions in Korea originate from livestock manure management according to CAPSS. Transportation is also a significant source in urban areas (T. Park et al., 2023). Highest VCDs are found in the southern SMA, where livestock farming is concentrated, and corresponding to a $PM_{2.5}$ hotspot (Figure 8b). Despite high $NH_3$ emissions in the southeast coast, VCD enhancements are not observed there due to high $SO_2$ emissions (Figure 3a) and expected high $SO_4^{2-}$ production converting gas-phase $NH_3$ to particle-phase ammonium ($NH_4^+$). Figure 9d indicates that annual total $NH_3$ emissions have shown little change while $NH_3$ VCDs have significantly increased since 2015. Decreases in $SO_2$ emissions and the resulting $SO_4^{2-}$ in both Korea and China have left more $NH_3$ available for $NO_3^-$ formation (J. Jeong et al., 2022).

Figure 9c shows $NO_3^-$ sensitivity regimes inferred from GEMS $NO_2$ and IASI $NH_3$ VCDs during the cold season (October−March) in 2023, as diagnosed using the winter threshold from Dang et al. (2024). Figure 9e shows the evolution of the sensitivity regimes inferred from OMI $NO_2$ and IASI $NH_3$ from 2015 to 2023. As $NO_x$ emissions have decreased, we find that $NO_3^-$ formation over Korea has transited from an $NH_3$-sensitive to a $NO_x$-sensitive regime. $NH_3$-sensitive conditions are now largely limited to parts of the SMA, and as $NO_x$ emissions continue to decrease we can expect $NO_3^-$ formation to be controlled by $NO_x$ emissions everywhere. Our analysis indicates that Korea will increasingly benefit from controlling $NO_x$ emissions to improve both $O_3$ and $PM_{2.5}$ air quality in the future.

## 4. Conclusions

We analyzed the distributions and 2015–2023 trends of major air pollutants in South Korea using the AirKorea surface network and satellite observations. Air quality in Korea has improved for primary pollutants over the past two decades, but surface $O_3$ and $PM_{2.5}$ still widely exceed national standards despite emission controls.

Surface CO and $SO_2$ levels have stayed below air quality standards since the late 1990s, while $NO_2$ is now below the air quality standard at almost all AirKorea sites. Anthropogenic CO and $SO_2$ show steady and consistent declines from 2015 to 2023 in both surface concentrations and satellite vertical column densities (VCDs), consistent with the trends from the CAPSS national emissions inventory. $NO_2$ surface concentrations decreased by 32% from 2015 to 2023 while tropospheric $NO_2$ VCDs decreased by 36%, consistent with the 23% decrease of $NO_x$ emissions in CAPSS.

Anthropogenic VOCs emissions, including a major contribution from aromatic compounds (BTEX), show no significant trend from 2015 to 2023 in the CAPSS inventory. This is consistent with BTEX observations at AirKorea sites and with HCHO and CHOCHO VCDs from satellites. Satellite HCHO observations show contributions from both anthropogenic and biogenic VOCs, while CHOCHO is more specifically associated with BTEX. Diurnal variations of HCHO and CHOCHO over the Seoul Metropolitan Area (SMA) observed from the GEMS geostationary satellite instrument show a morning increase and a leveling off by midday. Aircraft vertical columns over the SMA during the KORUS-AQ campaign show similar diurnal variations but a late afternoon HCHO increase.

Surface $O_3$ levels in Korea peak in May–June, and observations at AirKorea sites show an average increase of 0.8 ppbv $a^{-1}$ in $90^{th}$ percentile MDA8 $O_3$ from 2015 to 2023. Such an $O_3$ increase has been attributed to the effect of $NO_x$ emission reductions under VOC-sensitive conditions for $O_3$ production. However, we find from the evolution of the satellite $HCHO/NO_2$ ratio from 2015 to 2023 that the $O_3$ formation regime in Korea has been shifting from VOC- to $NO_x$-sensitive. GEMS satellite observations for 2023 indicate that most regions in Korea are now $NO_x$-sensitive or in a transition regime, and that VOC-sensitive conditions are confined to the central SMA. We find that the $O_3$ increase at AirKorea sites is limited to sites still in the VOC-sensitive regime, whereas there is no $O_3$ increase for sites in the transition or $NO_x$-limited regimes. Our results suggest that $O_3$ across Korea is poised to decrease in response to continued $NO_x$ emission controls.

Annual trends during 2015−2023 in $PM_{10}$, $PM_{2.5}$, and AOD show consistent decreases of 4−5% $a^{-1}$. Diurnal variations in AODs seen from the GOCI satellite instrument show the importance of photochemical production as a source of PM. The only $PM_{2.5}$ component not to show a significant decrease over the 2015−2023 period is nitrate ($NO_3^-$). From the $NH_3$/$NO_2$ ratio observed by satellites and its trend over the 2015−2023 period, we find that $PM_{2.5}$ $NO_3^-$ formation in Korea was mostly $NH_3$-sensitive but has become increasingly $NO_x$-sensitive as $NO_x$ emissions have decreased. As of 2023, $NO_3^-$ formation across Korea is dominantly $NO_x$-sensitive except in parts of the SMA.

The vigorous $NO_x$ emission controls in Korea starting in 2016 have not yet yielded results for decreasing $O_3$ and $PM_{2.5}$ $NO_3^-$. However, our results show that they have effectively shifted $O_3$ production from a VOC-sensitive to a $NO_x$-sensitive regime and $NO_3^-$ formation from an $NH_3$-sensitive to a $NO_x$-sensitive regime. As $NO_x$ emissions continue to decrease, the benefits for decreasing $O_3$ and $PM_{2.5}$ should become apparent.

**Acknowledgement**

This research has been supported by the Samsung Advanced Institute of Technology (grant no. A41602).

**Data availability**

AirKorea surface network data are available at https://www.airkorea.or.kr/eng. CAPSS annual emissions are available at https://www.air.go.kr/eng/main.do. KORUS-AQ aircraft data are available at https://www-air.larc.nasa.gov/cgi-bin/ArcView/korusaq. Satellite products are available at MOPITT CO https://l5ftl01.larc.nasa.gov:22000/misrl2l3/MOPITT/MOP03J.009/; OMI $SO_2$ https://dx.doi.org/10.5067/Aura/OMI/DATA3008, $NO_2$ https://dx.doi.org/10.5067/Aura/OMI/DATA3007, HCHO https://dx.doi.org/10.5067/Aura/OMI/DATA3010, CHOCHO https://doi.org/10.7910/DVN/Q1O2UE; TROPOMI CO https://dx.doi.org/10.5270/S5P-bj3nry0, $NO_2$ https://dx.doi.org/10.5270/S5P-9bnp8q8, HCHO https://dx.doi.org/10.5270/S5P-vg1i7t0; IASI $NH_3$ https://iasi.aeris-data.fr/nh3/; GEMS $SO_2$, HCHO, CHOCHO https://nesc.nier.go.kr/en/html/index.do, $NO_2$ https://doi.org/10.7910/DVN/ZQQJRO; GOCI AOD available upon request.

**Author contributions**

Original draft preparation, data processing, analysis, investigation, and visualization were done by YJO. DJJ contributed to project conceptualization. Review and editing were done by DJJ, DCP, RD, HC, SL, and JK. DCP, NKC, and SK provided additional resources and support in analysis.

## Competing interests

The contact author has declared that none of the authors has any competing interests.

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

**Figures**

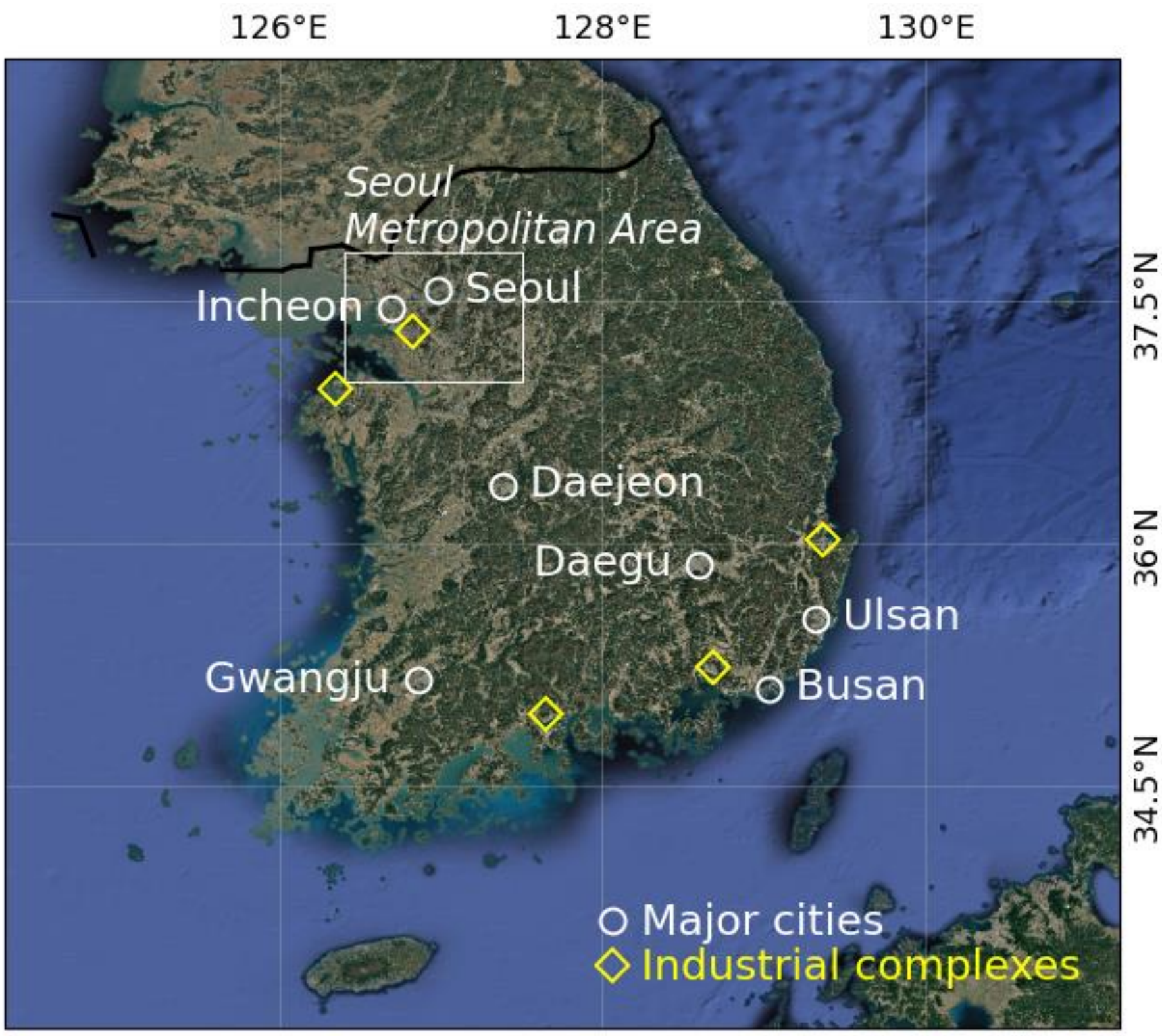

**Figure 1. Geographic locations of major source regions in South Korea.** Major cities and industrial complexes are indicated in white and yellow colors. The Seoul Metropolitan Area (SMA) is defined as the rectangular domain covering 37–37.8° N and 126.4–127.5° E. Background surface imagery is from © Google Earth.

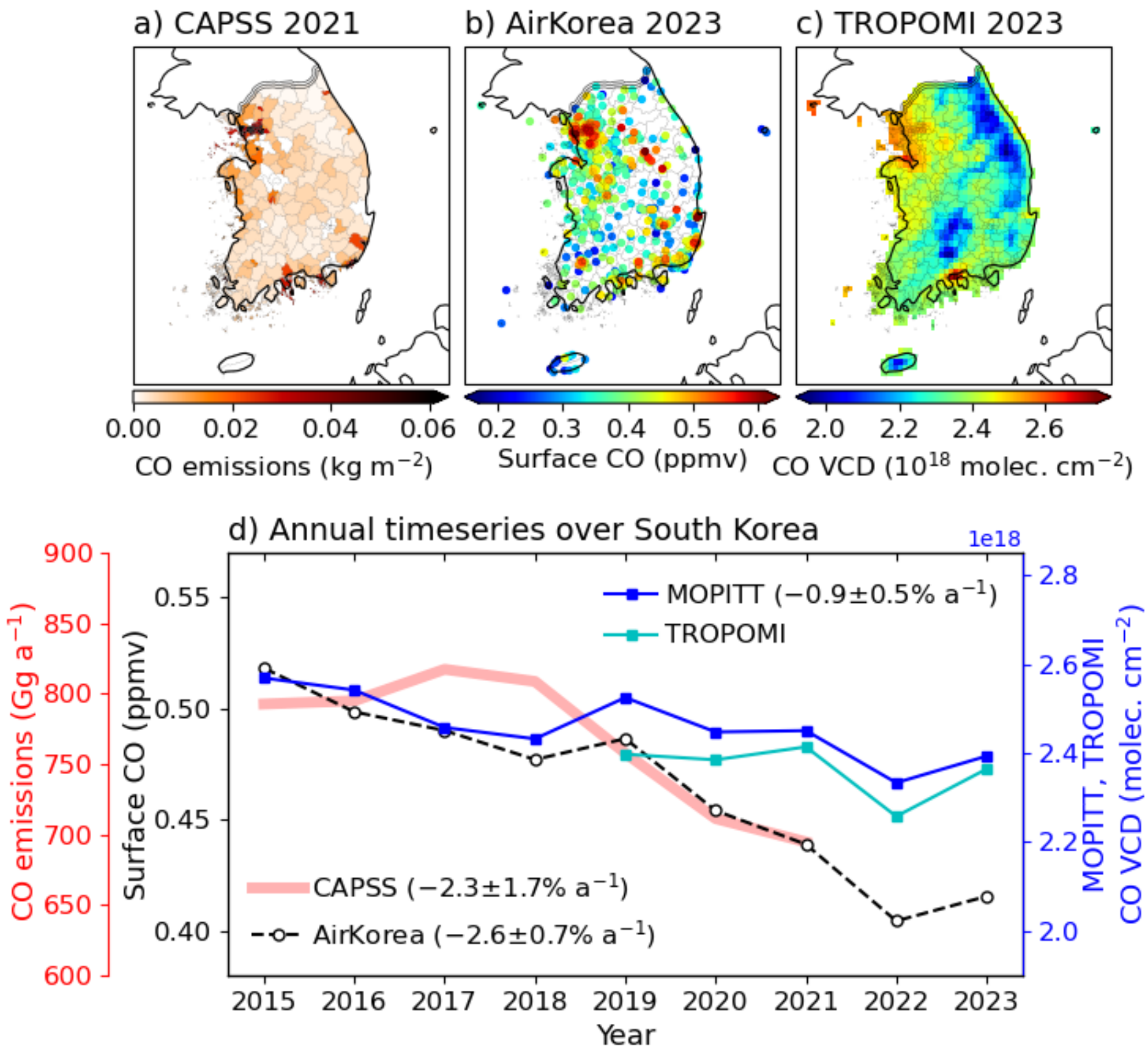

**Figure 2. Annual mean CO distributions and trends in South Korea.** Top panels show spatial distributions of (a) 2021 anthropogenic CO emissions from CAPSS, (b−c) 2023 average AirKorea surface CO concentrations and TROPOMI CO vertical column densities (VCDs). VCDs are mapped on a 0.1° × 0.1° grid. Lower panel (d) shows 2015−2023 trends in CAPSS CO emissions, surface CO averaged over all AirKorea sites, and CO VCDs from TROPOMI and MOPITT averaged over South Korea. Statistically significant trends ($p$-value < 0.05) are given inset.

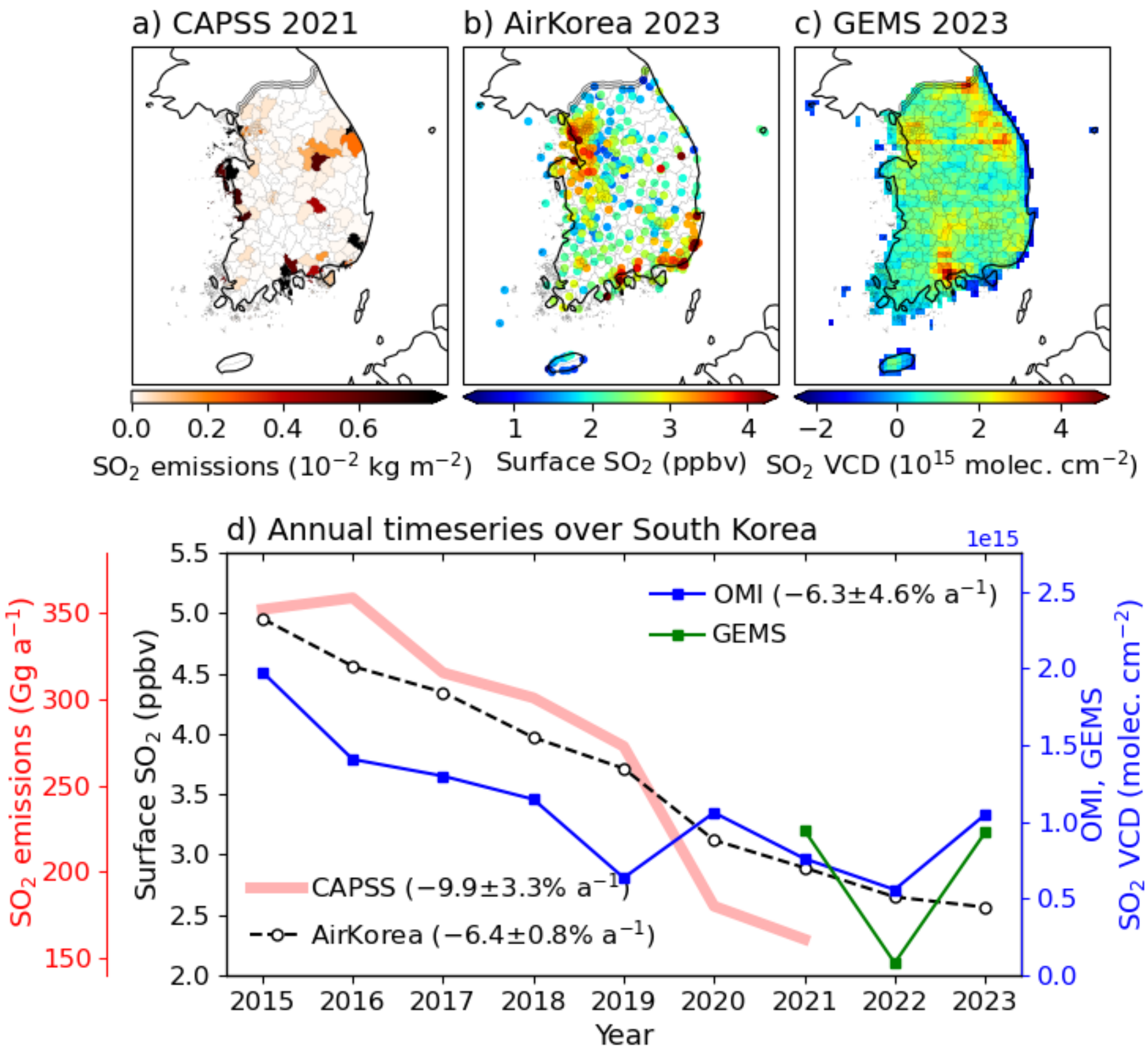

**Figure 3. Annual mean $SO_2$ distributions and trends in South Korea.** Top panels show spatial distributions of (a) 2021 anthropogenic $SO_2$ emissions from CAPSS, (b−c) 2023 average AirKorea surface $SO_2$ concentrations and GEMS $SO_2$ VCDs. VCDs are mapped on a 0.1° × 0.1° grid. Lower panel (d) shows 2015−2023 trends in CAPSS $SO_2$ emissions, surface $SO_2$ averaged over all AirKorea sites, and $SO_2$ VCDs from OMI and GEMS (sampled at OMI overpass time) averaged over South Korea. Statistically significant trends (*p*-value < 0.05) are given inset.

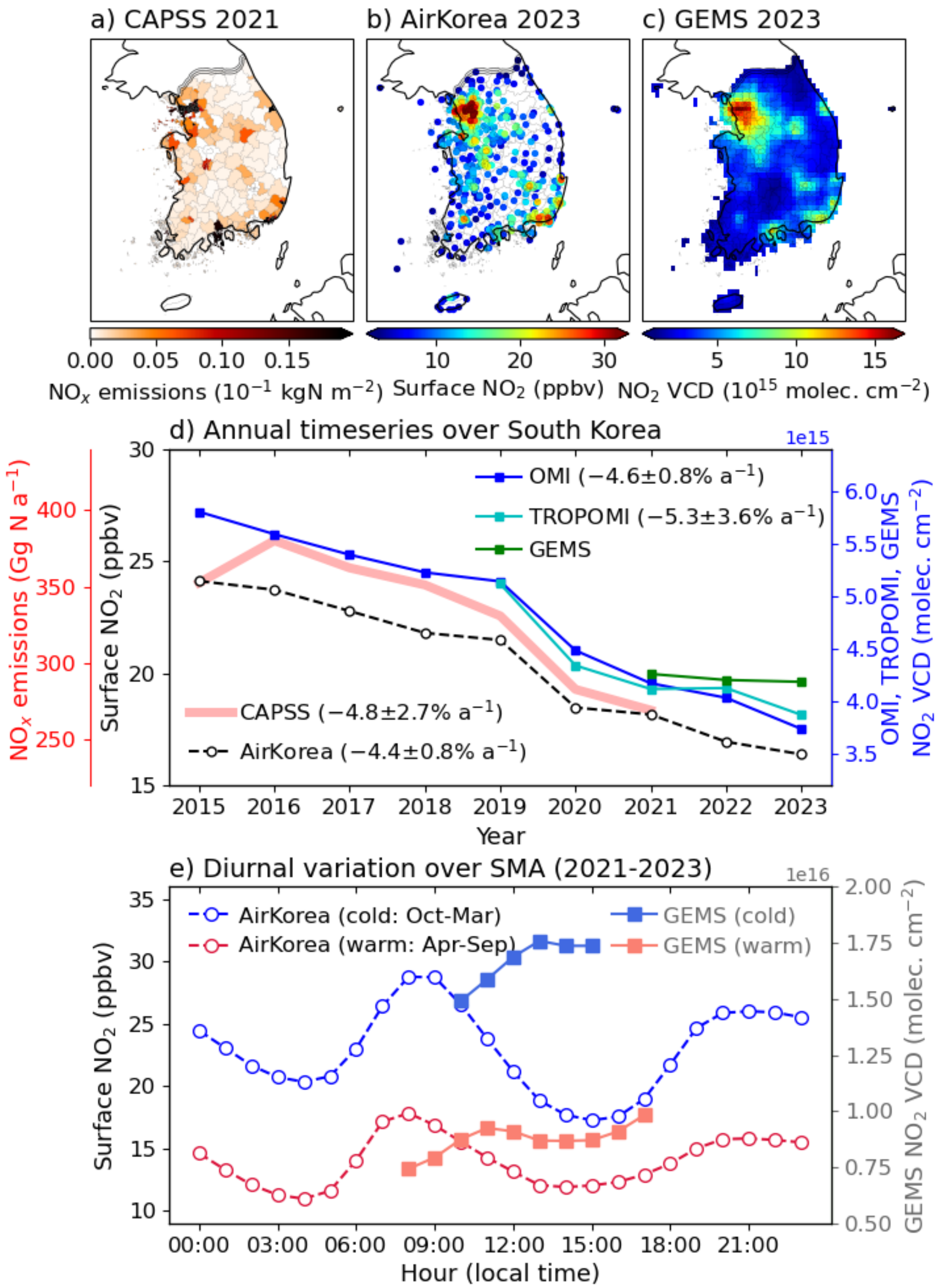

**Figure 4. Annual mean $NO_2$ distributions and trends in South Korea.** Top panels show spatial distributions of (a) 2021 anthropogenic $NO_x$ emissions from CAPSS, (b–c) 2023 average AirKorea surface $NO_2$ concentrations and GEMS tropospheric $NO_2$ VCDs. VCDs are mapped on a 0.1° × 0.1° grid.

Middle panel (d) shows 2015−2023 trends in CAPSS $NO_x$ emissions, surface $NO_2$ averaged over all AirKorea sites, and tropospheric $NO_2$ VCDs from OMI, TROPOMI, and GEMS (sampled at OMI overpass time) averaged over South Korea. Statistically significant trends ($p$-value < 0.05) are given inset. Lower panel (e) shows 2021−2023 seasonal mean (cold: October−March, warm: April−September) diurnal variations of AirKorea surface $NO_2$ concentrations and GEMS VCDs in the SMA.

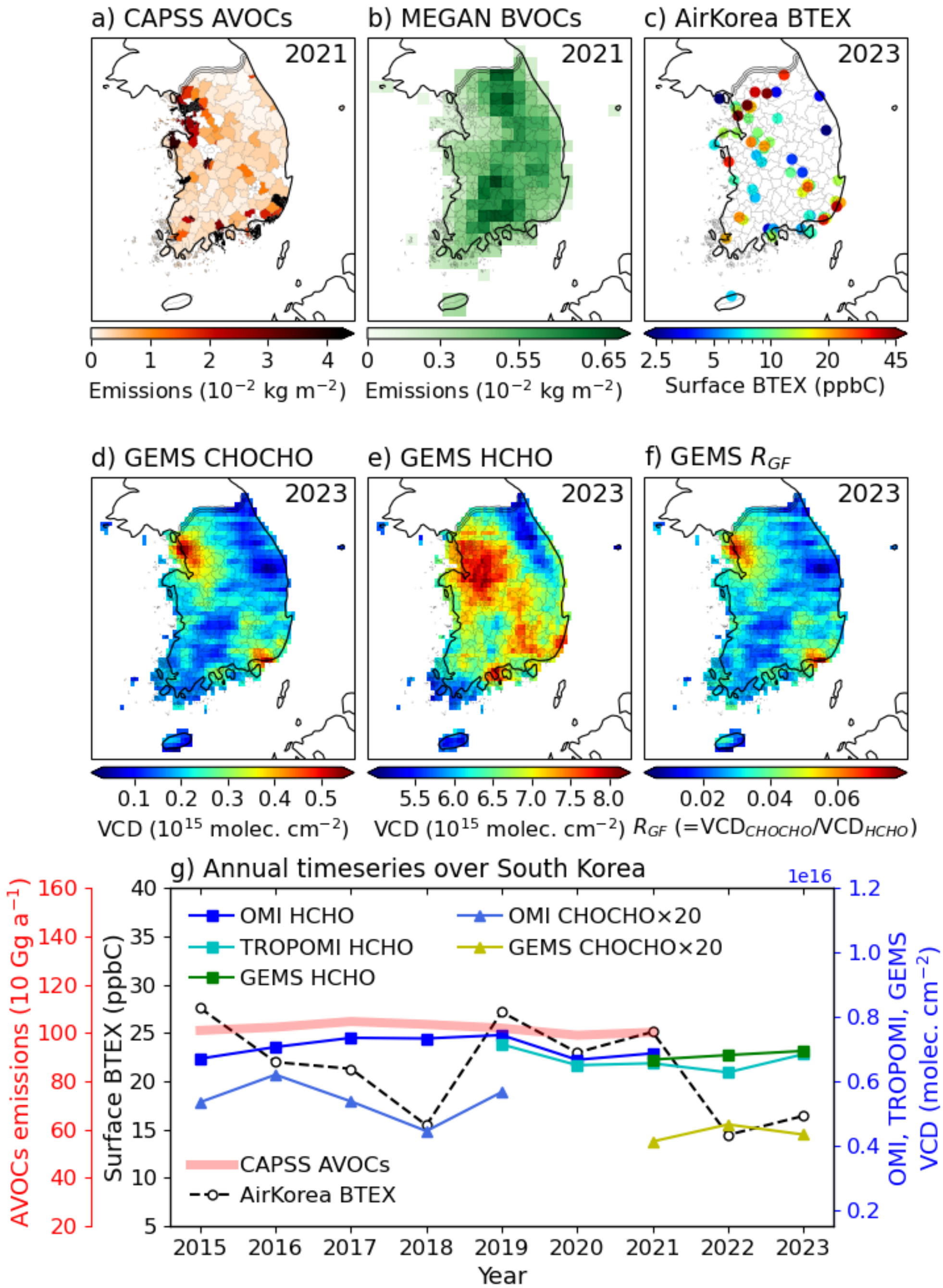

**Figure 5. Annual mean NMVOC distributions and trends in South Korea.** Top panels (a–b) show 2021 anthropogenic VOCs (AVOCs) emissions from CAPSS and biogenic VOCs (BVOCs: sum of isoprene, monoterpenes, sesquiterpenes, acetaldehyde, acetone, methanol, ethanol) emissions from MEGAN, and (c) 2023 average AirKorea surface BTEX (≡ benzene + toluene + ethylbenzene + xylenes)

concentrations. Middle panels (d−f) show spatial distributions of 2023 average GEMS glyoxal (CHOCHO) VCDs, formaldehyde (HCHO) VCDs, and glyoxal to formaldehyde ratio $R_{GF}$ (= $VCD_{CHOCHO}/VCD_{HCHO}$) mapped on 0.1° × 0.1° grids. Lower panel (g) shows 2015−2023 trends in CAPSS AVOCs emissions, surface BTEX averaged over available AirKorea sites, and CHOCHO and HCHO VCDs from OMI, TROPOMI, and GEMS (sampled at OMI overpass time) averaged over South Korea. None of the data show significant trends over the 2015−2023 period.

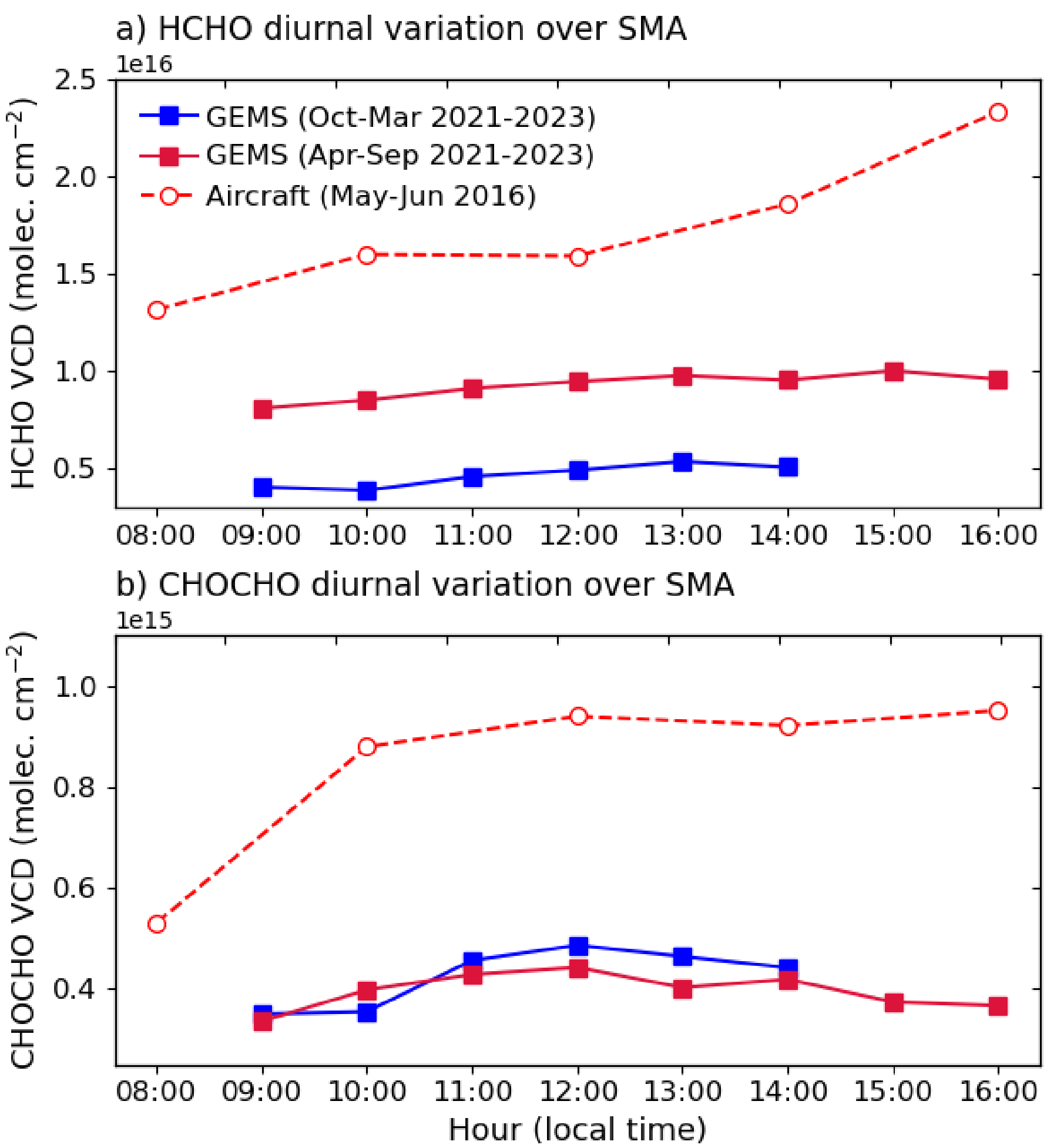

**Figure 6. Diurnal variations of HCHO and CHOCHO VCDs in the SMA.** Upper panel (a) shows seasonal mean (blue: October−March, red: April−September) diurnal variations of HCHO VCDs from GEMS 2021−2023 observations and KORUS-AQ (May−June 2016) DC-8 aircraft observations below 8 km altitude over the SMA. Lower panel (b) shows the same for CHOCHO VCDs.

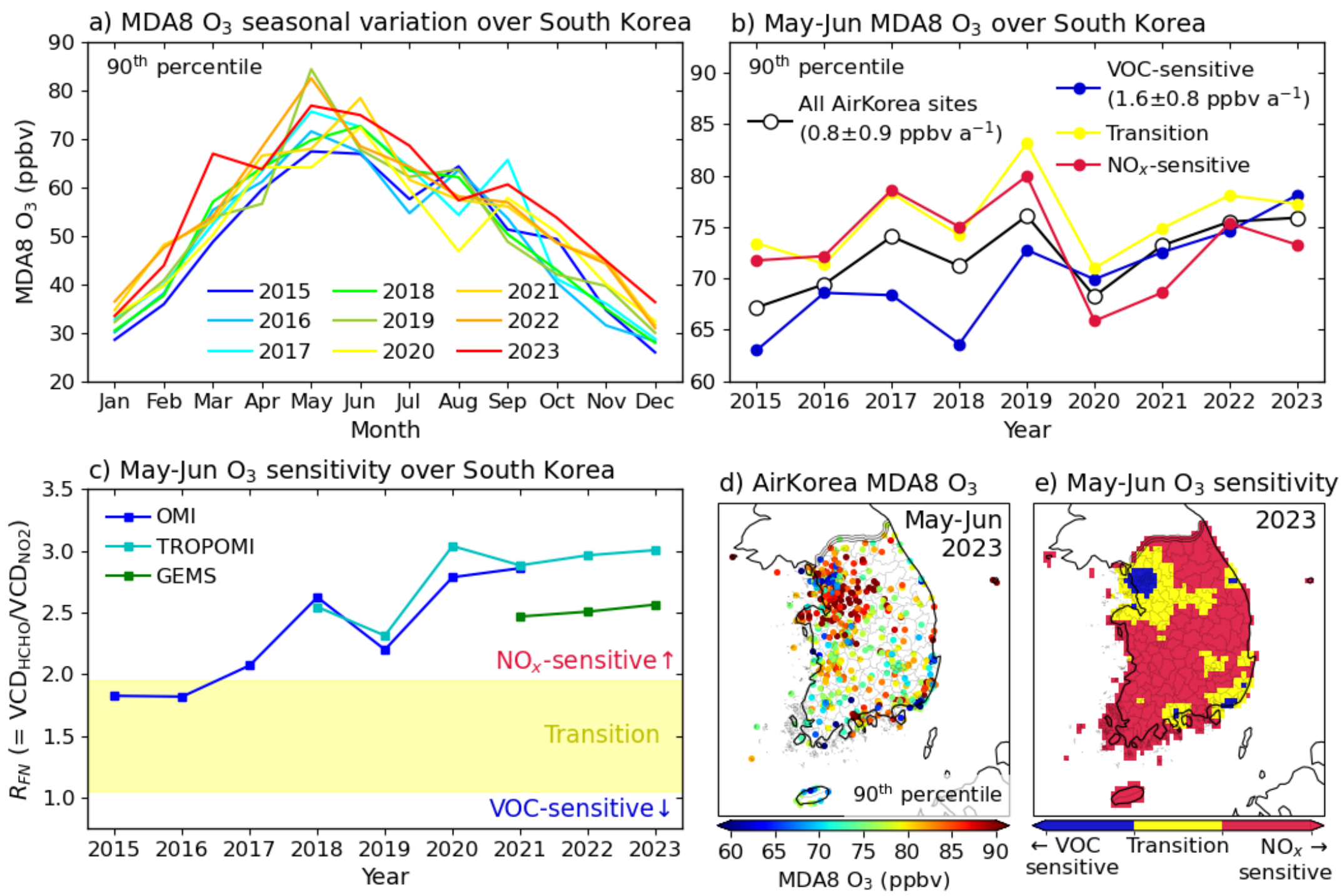

**Figure 7. $O_3$ distribution, trend, and sensitivity to precursors in South Korea.** Values are shown for the 90th percentile maximum 8-hour daily average (MDA8) at individual AirKorea sites. Top panels show averages of 90th percentile MDA8 $O_3$ for 2015–2023 as (a) monthly variations in individual years and (b) long-term trends in May–June (when concentrations are highest) for sites in different sensitivity regimes inferred from 2023 GEMS observations. Statistically significant trends ($p$-value < 0.05) are given inset. Lower left panel (c) shows May–June average timeseries of formaldehyde to $NO_2$ ratios $R_{FN}$ (= $VCD_{HCHO}/VCD_{NO2}$) from OMI, TROPOMI, and GEMS (sampled at OMI overpass time). Lower right panels show spatial distributions of May–June 2023 average (d) AirKorea 90th percentile MDA8 $O_3$ and (e) $O_3$ sensitivity regimes inferred from GEMS $R_{FN}$ mapped on a 0.1° × 0.1° grid. $O_3$ sensitivity regimes are based on $R_{FN}$ thresholds from Duncan et al. (2010).

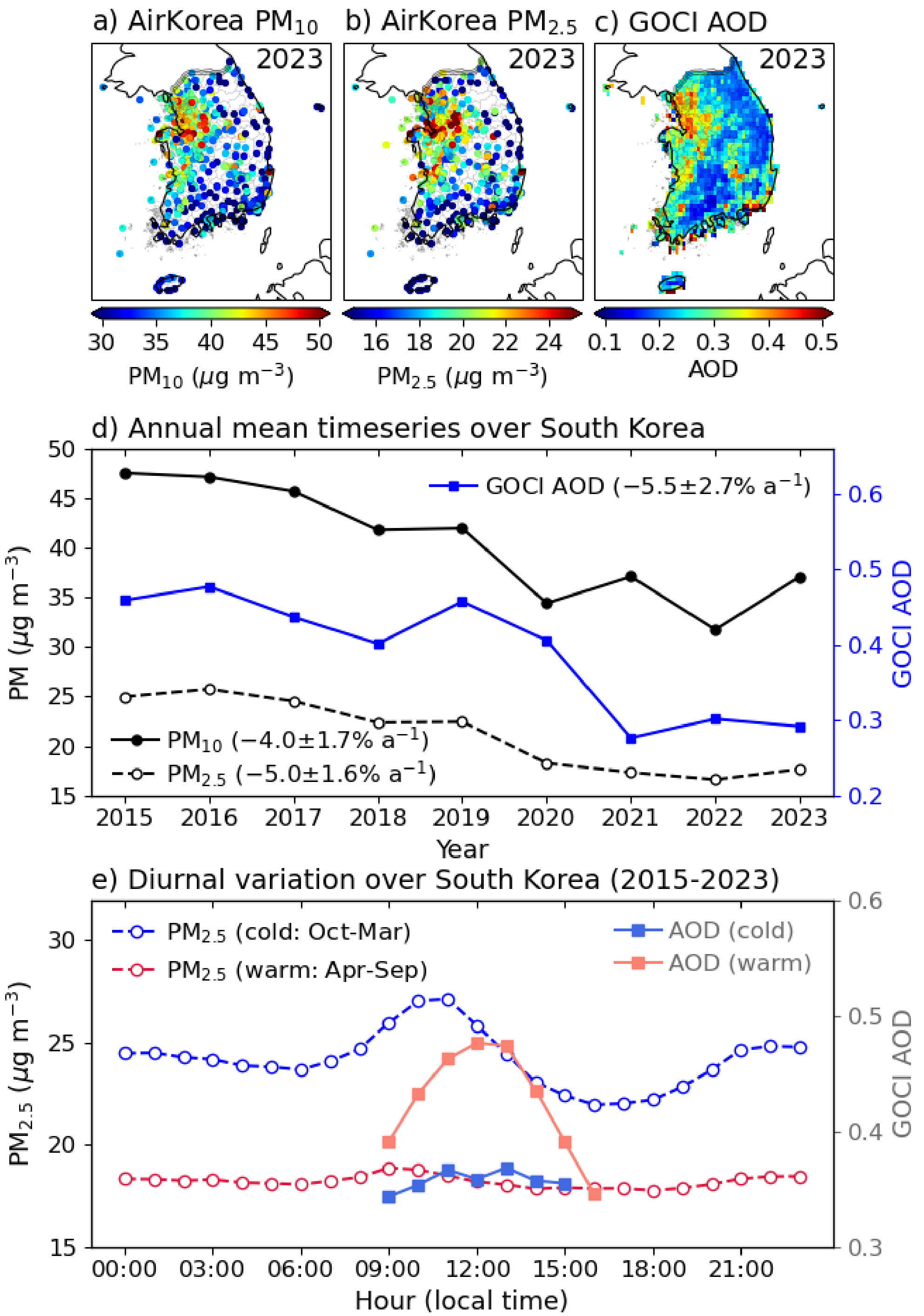

**Figure 8. Annual mean PM and aerosol optical depth (AOD) distributions and trends in South Korea.** Top panels (a–c) show spatial distributions of 2023 average AirKorea $PM_{10}$ and $PM_{2.5}$, as well as

GOCI (GOCI; 2015–2020, GOCI-II; 2021–2023) AOD. AOD is mapped on a 0.1° × 0.1° grid. Middle panel (d) shows 2015−2023 trends in $PM_{10}$ and $PM_{2.5}$ averaged over all AirKorea sites, and GOCI AOD averaged over South Korea. Statistically significant trends (*p*-value < 0.05) are given inset. Lower panel (e) shows 2015−2023 seasonal mean (cold: October−March, warm: April−September) diurnal variations of AirKorea $PM_{2.5}$ concentrations and GOCI AOD over South Korea.

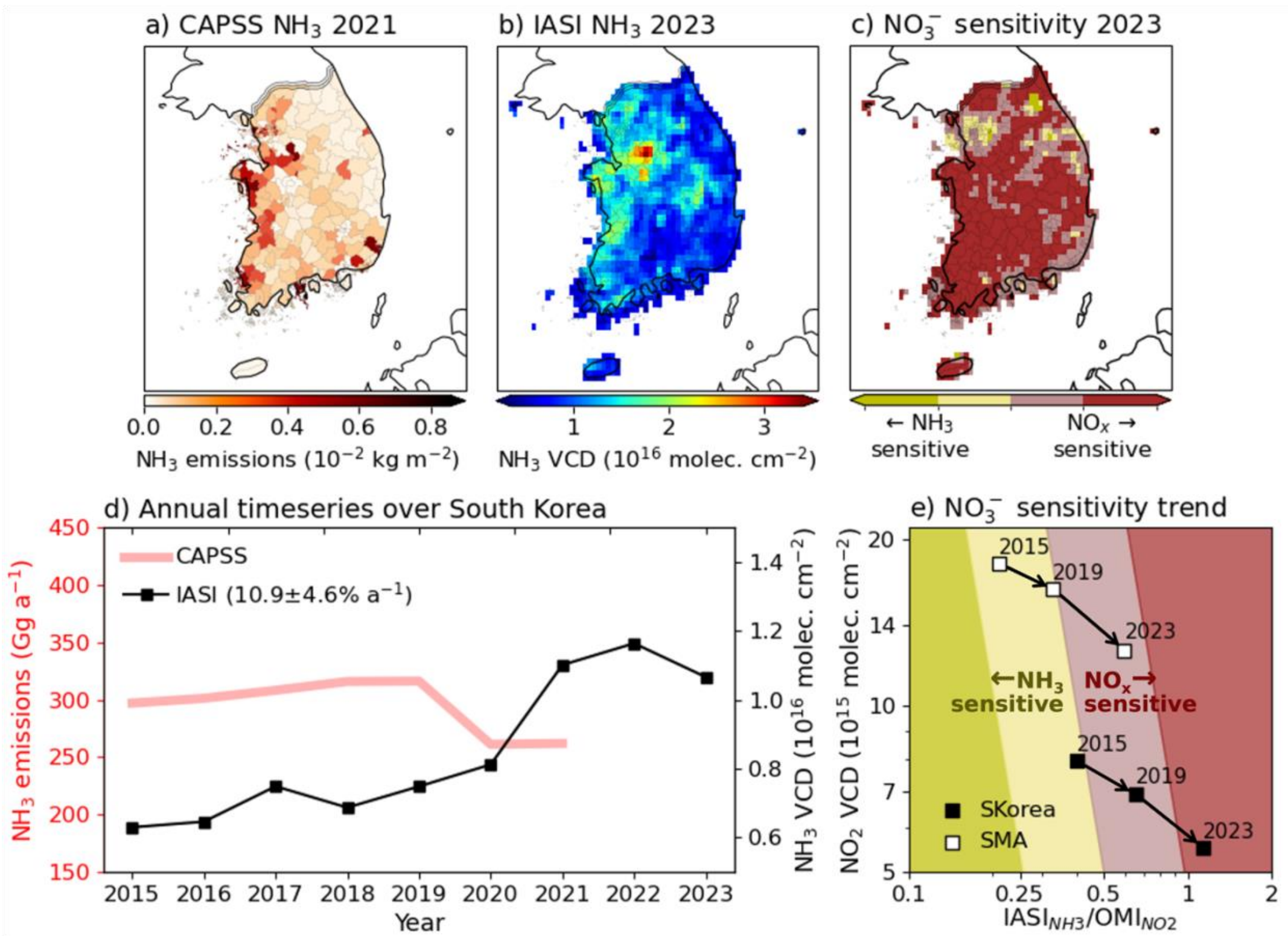

**Figure 9. Annual mean $NH_3$ distributions, trends, and $PM_{2.5}$ nitrate ($NO_3^-$) sensitivity in South Korea.** Top panels show spatial distributions of (a) 2021 anthropogenic $NH_3$ emissions from CAPSS, (b) 2023 average IASI $NH_3$ VCDs, and (c) 2023 cold season (October−March) $NO_3^-$ sensitivity regimes inferred from IASI $NH_3$ and GEMS $NO_2$. VCDs are mapped on a 0.1° × 0.1° grid. Lower panel (d) shows 2015−2023 trends in CAPSS $NH_3$ emissions and IASI $NH_3$ VCDs averaged over South Korea. Statistically significant trends (*p*-value < 0.05) are given inset. Lower right panel (e) shows the cold season $NO_3^-$ sensitivity trends averaged over South Korea and over the SMA. $NO_3^-$ sensitivity regimes are based on winter thresholds from Dang et al. (2024).