# Peer review of "Air quality trends and regimes in South Korea inferred from 2015–2023 surface and satellite observations"

_EGUsphere, 2024_

## Referee Comment (RC1)

This manuscript investigates 2015−2023 trends of major air pollutants in South Korea using the AirKorea surface network and satellite observations. The results indicate that further $NO_x$ emission decreases in Korea will reap benefits for both $O_3$ and $PM_{2.5}$ pollution. The figures are well prepared, and the analyses are relatively sound based on the results. This research's quality and scope are suitable for publication in ACP. However, the manuscript still requires revision to ensure a high-quality analysis that meets ACP standards, subject to the following concerns.

Major Comments:

1. The manuscript analyses trends in major air pollution in Korea using multi-source data, including satellites, ground-based observations, and emission inventories…, but there are significant differences in trends between the multi-source data that need to be clarified. (a) Authors claim CO trend observed by MOPITT decreased slower than surface concentrations because of the background contribution to the CO VCD (Line155-156), why there is a consistent downward trend in MOPITT and surface concentrations in the period 2015-2018 and a huge difference in their downward trends in 2019-2023, it is clear that there is more than just the effect of background concentrations here. (b) Surface $SO_2$ concentrations and OMI VCDs have decreased at similar rates but there are differences (Line173), For example, $SO_2$ observed by OMI rises significantly in 2019-2020, and $SO_2$ observed by both OMI and GEMS rises in 2022-2023, whereas CAPSS and AirKorea only show a downward trend, and these details should be clarified. (c) I really don't understand why the diurnal variation of $NO_2$ VCD observed by GEMS in warm season (8-11:00 local time) and cold season (10-13:00 local time) is opposite to that of surface $NO_2$ (Figure 4e). The authors try to explain this phenomenon by using the variation of the mixed layer height, it is insufficient. Besides, the high $NO_2$ concentration in the morning and evening is affected by meteorological conditions. Vehicle emissions during the morning and evening rush hours are also an important factor. $NO_2$ is mainly concentrated near the surface and rapidly photolysis after sunrise, and the satellite and the surface observations should show similar diurnal trends, which can be confirmed by previous observations in some mega-cities (Tian et al., 2018) and background stations (Cheng et al., 2019). What's more, an observation from the GEMS also showed that $NO_2$ column concentrations began to decline at 10:00 (local time) (Xu et al., 2023). I recommend first comparing the GEMS and surface $NO_2$ concentrations on an hour-by-hour basis, and then carefully analysing the reasons for the opposite trend.

2. Line288-289 "Based on the criteria from Duncan et al. (2010) the positive trend in RFN implies that Korea is now mostly in the NOx-sensitive regime (RFN > 2)." In order to avoid the misjudgment of $O_3$ formation sensitivity caused by arbitrary selection of FNR thresholds, I strongly suggest using a third–order polynomial model to investigate the empirical relationship between FNR and surface $O_3$ concentrations, which has been widely used in other studies (Ren et al., 2022; Jin et al., 2020). The criteria presented in Duncan et al. (2010) may not be applicable to the current diagnosis of $O_3$ formation sensitivity, the threshold is usually small (1 and 2), which causes the contribution of the NOx limit regime to be overestimated.

Minor Comments:

1. Line 57-58 "Synoptic meteorology and transport from China also contribute to seasonal and long-term variations of pollutants over Korea." Missing relevant references.

2. Line 204-205 "Both surface and column $NO_2$ are higher by a factor of two during the cold season, which can be explained by the longer $NO_x$ lifetime (Shah et al., 2020)." Differences in warm- and cold-season emission patterns should have a greater impact.

3. Line 242-243 "but CHOCHO shows hotspots for manufacturing industries while HCHO shows hotspots for petrochemical facilities." Unclear HCHO shows hotspots for petrochemical facilities, since HCHO observations are also more distributed, HCHO didn't just indicate petrochemical facilities.

4. Line 262-263 "has been previously reported as systematic low biases in satellite observations of CHOCHO and HCHO." Please specify it.

5. GEMS is observed every hour during the day and the time should be clarified. For example, in Fig. 3d, does GEMS use all the observations during the day or just a certain hour of the mid-day.

6. Figure 5g "OMI CHOCHO×20", Does it mean 20 times magnification? This should be clarified in the legend.

Suggestion:

Although well known, some instrument name abbreviations should indicate the full name when they first appear, i.e. OMI, TROPOMI, MOPITT…

Reference:

Cheng S, Ma J, Cheng W, et al. Tropospheric NO2 vertical column densities retrieved from ground-based MAX-DOAS measurements at Shangdianzi regional atmospheric background station in China[J]. Journal of Environmental Sciences, 2019, 80: 186-196.

Tian X, Xie P, Xu J, et al. Long-term observations of tropospheric NO2, SO2 and HCHO by MAX-DOAS in Yangtze River Delta area, China[J]. Journal of Environmental Sciences, 2018, 71: 207-221.

Xu T, Zhang C, Xue J, et al. Estimating hourly nitrogen oxide emissions over East Asia from geostationary satellite measurements[J]. Environmental Science & Technology Letters, 2023, 11(2): 122-129.

Ren J, Guo F, Xie S. Diagnosing ozone–NO x–VOC sensitivity and revealing causes of ozone increases in China based on 2013–2021 satellite retrievals[J]. Atmospheric Chemistry and Physics, 2022, 22(22): 15035-15047.

Jin X, Fiore A, Boersma K F, et al. Inferring changes in summertime surface Ozone–NO x– VOC chemistry over US urban areas from two decades of satellite and ground-based observations[J]. Environmental science & technology, 2020, 54(11): 6518-6529.

---

## Author Comment (AC1)

We thank the reviewers for their thorough evaluation and suggestions. We have carefully addressed the comments below.

**RC1**

**Major Comments**:

1.  The manuscript analyses trends in major air pollution in Korea using multi-source data, including satellites, ground-based observations, and emission inventories…, but there are significant differences in trends between the multi-source data that need to be clarified.

    (a) Authors claim CO trend observed by MOPITT decreased slower than surface concentrations because of the background contribution to the CO VCD (Line155-156), why there is a consistent downward trend in MOPITT and surface concentrations in the period 2015-2018 and a huge difference in their downward trends in 2019-2023, it is clear that there is more than just the effect of background concentrations here.

    We do not consider the difference to be huge, and its cause is not clear so we would rather not discuss it. We specified the background CO for clarity (Lines 170-172).

    *"MOPITT decreases at a rate of $-0.9 \pm 0.5\%$ $a^{-1}$, slower than surface concentrations because of the background contribution to the VCDs ($\sim 2 \times 10^{18}$ molecules $cm^{-2}$)."*

    (b) Surface $SO_2$ concentrations and OMI VCDs have decreased at similar rates but there are differences (Line173), For example, $SO_2$ observed by OMI rises significantly in 2019-2020, and $SO_2$ observed by both OMI and GEMS rises in 2022-2023, whereas CAPSS and AirKorea only show a downward trend, and these details should be clarified.

    The retrieval uncertainties of $SO_2$ from OMI and GEMS are large, therefore interannual variations appear to be associated with random or systematic errors during the retrieval. We clarified in the text as follows (Lines 188-189):

    *"There is large uncertainty in the satellite observations that likely contributes noise to the trend (J. Kim et al., 2020; C. Li et al., 2020)."*

    (c) I really don't understand why the diurnal variation of $NO_2$ VCD observed by GEMS in warm season (8-11:00 local time) and cold season (10-13:00 local time) is opposite to that of surface $NO_2$ (Figure 4e). The authors try to explain this phenomenon by using the variation of the mixed layer height, it is insufficient. Besides, the high $NO_2$ concentration in the morning and evening is affected by meteorological conditions. Vehicle emissions during the morning and evening rush hours are also an important factor. $NO_2$ is mainly concentrated near the surface and rapidly photolysis after sunrise, and the satellite and the surface observations should show similar diurnal trends, which can be confirmed by previous observations in some mega-cities (Tian et al., 2018) and background stations (Cheng et al., 2019). What's more, an observation from the GEMS also showed that $NO_2$ column concentrations began to decline at 10:00 (local time) (Xu et al., 2023). I recommend first comparing the GEMS and surface $NO_2$ concentrations on an hour-by-hour basis, and then carefully analysing the reasons for the opposite trend.

    We mainly attribute the surface diurnal variability to the mixed layer height growth because traffic load in Seoul has weak daytime variability resulting in relatively flat emissions instead of a bimodal behavior (Yang et al., 2024). As pointed out, the photochemical sink plays a role in the daytime minimum, so we revised the explanation as follows (Lines 218-229):

*"NO$_x$ emissions in the SMA have small seasonal variations as they are dominated by mobile sources (Pandey et al., 2008; H. M. Lee and R. Park, 2022; Yang et al., 2024). The emissions are higher in the daytime (7−18 LT) than at night but do not show significant rush hour enhancements because traffic load is sustained with little variability throughout the daytime (Yang et al., 2024). Therefore, the peak in surface NO$_2$ concentrations at 8−9 LT is not due to the rush hour but to accumulation of daytime emissions in a shallow mixed layer (Moutinho et al., 2020). NO$_2$ then decreases in the morning by dilution as the mixed layer grows from solar heating, and increases again in the evening when the mixed layer collapses (J. Li et al., 2021). Increasing NO$_2$ photolysis as the morning progresses would also be expected to lower NO$_2$ concentrations but this is offset by entrainment of O$_3$ from aloft as the mixed layer grows, such that the NO$_2$/NO$_x$ ratio increases during the morning hours (Yang et al., 2024)."*
We find an 11 LT peak and subsequent decline in NO$_2$ VCDs during Apr-Sep, which is consistent with the 10-11 LT peak in Seoul during May-Sep from Xu et al. (2024). However, as pointed out by Xu et al. (2024) and shown in the comparison across major Chinese cities by Tian et al. (2018), diurnal patterns of VCDs vary depending on emission characteristics (energy consumption and transportation). There is also a clear difference in background regions as shown by Cheng et al. (2019), where the VCDs do not show morning accumulation of emissions but only a 11-14 LT minimum caused by the dominance of chemical loss and ventilation. We elaborated in more detail as follows (Lines 229-234):
*"Geostationary satellite observations provide unique information on the diurnal variation of NO$_2$ VCDs (Tian et al., 2018; Cheng et al., 2019; Edwards et al., 2024; Xu et al., 2024). This is illustrated in Figure 4e for the SMA. A NO$_x$ budget analysis by Yang et al. (2024) shows that NO$_2$ VCDs in Seoul increase steadily in the morning from accumulation of emissions as they are not affected by mixed layer growth, reaching a steady state in the afternoon due mostly to loss from ventilation."*

2. Line288-289 "Based on the criteria from Duncan et al. (2010) the positive trend in RFN implies that Korea is now mostly in the NOx-sensitive regime (RFN > 2)." In order to avoid the misjudgment of O$_3$ formation sensitivity caused by arbitrary selection of FNR thresholds, I strongly suggest using a third–order polynomial model to investigate the empirical relationship between FNR and surface O$_3$ concentrations, which has been widely used in other studies (Ren et al., 2022; Jin et al., 2020). The criteria presented in Duncan et al. (2010) may not be applicable to the current diagnosis of O$_3$ formation sensitivity, the threshold is usually small (1 and 2), which causes the contribution of the NOx limit regime to be overestimated.

We agree that the $R_{FN}$ threshold may vary with time and region. Recent analyses by the authors of Jin et al. (2020) used GEOS-Chem to derive region-specific thresholds across the globe and applied the criteria to GEMS observations (Jin et al., 2024). Their results imply that $R_{FN} > 2\sim3$ is a reasonable threshold applicable to South Korea, similar to the results over China (2.2-3.2) by Ren et al. (2022). We replotted Figure 7 using a threshold of 2.5 and revised the manuscript as follows (Lines 309-314):
*"Recent studies over Northeast Asia suggest that NO$_x$-limited regimes are found where $R_{FN} > 2–3$ (Ren et al., 2022; Jin et al., 2020, 2024). Here we use 2.5 as a threshold and find that Korea is now mostly in the NO$_x$-limited regime. Figure 7d–e shows May−June 2023 MDA8 O$_3$ and its sensitivity regimes inferred from GEMS $R_{FN}$. Most of the country is in a NO$_x$-limited and transition regimes while VOC-limited conditions are largely limited to the central SMA and Busan."*
**Reference:** Jin X., Yang, Y., and Wang, S.: Observing the diurnal cycle of ozone-NO$_x$-VOC sensitivity from geostationary satellite retrievals of ozone precursors, Abstract A21I-1872 presented at AGU24, Washington DC, 9-13 December, 2024

**Minor Comments:**

1. Line 57-58 "Synoptic meteorology and transport from China also contribute to seasonal and long-term variations of pollutants over Korea." Missing relevant references.
   Revised as follows:
   *"Synoptic meteorology and transport from China also contribute to seasonal and long-term variations of pollutants over Korea (H. M. Lee and R. Park, 2022; D. Park et al., 2021; J. Jeong et al., 2024)."*

2. Line 204-205 "Both surface and column $NO_2$ are higher by a factor of two during the cold season, which can be explained by the longer $NO_x$ lifetime (Shah et al., 2020)." Differences in warm- and cold-season emission patterns should have a greater impact.
   Several studies confirm that $NO_x$ emissions in the SMA are mainly from mobile sources which are persistent throughout the year, with small seasonal variations. We clarified this as follows:
   *"$NO_x$ emissions in the SMA have small seasonal variations as they are dominated by mobile sources (Pandey et al., 2008; H. M. Lee and R. Park, 2022; Yang et al., 2024)."*

3. Line 242-243 "but CHOCHO shows hotspots for manufacturing industries while HCHO shows hotspots for petrochemical facilities." Unclear HCHO shows hotspots for petrochemical facilities, since HCHO observations are also more distributed, HCHO didn't just indicate petrochemical facilities.
   We notice a distinct contrast in the CHOCHO and HCHO hotspots over industrial regions with different facilities and clarified as follows:
   *"CHOCHO shows hotspots for manufacturing industries (Incheon, Changwon) while HCHO shows hotspots for petrochemical facilities (Yeosu, Ulsan)."*

4. Line 262-263 "has been previously reported as systematic low biases in satellite observations of CHOCHO and HCHO." Please specify it.
   Revised as follows:
   *"We find that the GEMS columns are lower than the aircraft columns, consistent with previously reported low biases in satellite retrievals of CHOCHO (−50%) and HCHO (−40% to −20%) (Chan Miller et al., 2017; Zhu et al., 2016; Zhu et al., 2020)."*

5. GEMS is observed every hour during the day and the time should be clarified. For example, in Fig. 3d, does GEMS use all the observations during the day or just a certain hour of the mid-day.
   We included detailed descriptions on satellite overpass times and how GEMS is sampled in the text (Lines 125-131) and figures.
   *"OMI and TROPOMI make afternoon overpasses at 13:30 local time (LT). We make use of morning overpasses for MOPITT (10:30 LT) and IASI (9:30 LT). We use hourly daytime observations from GEMS (7:45−16:45 LT), GOCI (9:30−16:30 LT), and GOCI-II (8:15−17:15 LT). For annual trend analyses we use GEMS observations made between 12−14 LT for consistency with the overpass time of OMI and TROPOMI measuring the same gases. We find no significant differences in observed trends when using surface observations sampled at satellite overpass times and therefore use all hours of the day."*

6. Figure 5g "OMI CHOCHOÍ20", Does it mean 20 times magnification? This should be clarified in the legend.
   We clarified this in the figure caption.

**Suggestion:**
Although well known, some instrument name abbreviations should indicate the full name when they first appear, i.e. OMI, TROPOMI, MOPITT…
   We spelled out full names in the introduction as suggested.

---

## Author Comment (AC2)

We thank the reviewers for their thorough evaluation and suggestions. We have carefully addressed the comments below.

**RC2**
**Specific comments:**

1. Line 16: Include a statement on the context/background of the research at the start of the abstract.

   We revised the abstract accordingly.

2. Line 59: Specify how the summer monsoon impacts $O_3$. The $O_3$ decreases after June due to the monsoon doing what? The current implication of the sentence is that the monsoon causes a peak in May-June $O_3$.

   We revised the sentence referring to the results from Lee and Park (2022) and Wie et al. (2018).
   *"Photochemical $O_3$ production is largest during the summer, but the summer monsoon brings clean marine air masses into the Korean peninsula resulting in lower $O_3$ levels in July–August compared to May–June (Wie et al., 2018; H. M. Lee and R. Park, 2022). May–June also has additional contributions to $O_3$ from wildfires, stratospheric intrusions, and transport from China (H. M. Lee and R. Park, 2022)."*

3. Lines 74-77: Restructure to first introduce $O_3$ sensitivity to VOCs versus $NO_x$ and then the relevant trace gas ratio. Similarly, for $NO_3^-$ You could also expand on why $O_3$ and $NO_3^-$ are sensitive to these specific compounds earlier in the paragraph.

   Revised as follows:
   *"Photochemical $O_3$ production takes place by oxidation of VOCs and CO in the presence of $NO_x$, and can be either $NO_x$- or VOC-limited depending on the concentrations of these precursors. Formation of $PM_{2.5}$ $NO_3^-$, which is a major component of wintertime secondary $PM_{2.5}$ in Korea and is mainly present as ammonium nitrate, can be either $NO_x$- or ammonia ($NH_3$)-sensitive again depending on the concentrations of these precursors. These dependences define chemical regimes that are important to identify for emission control strategies. $O_3$ sensitivity to $NO_x$ versus VOCs can be diagnosed using formaldehyde (HCHO) to $NO_2$ ratios measured from satellites, where HCHO and $NO_2$ are proxies for VOCs and $NO_x$ emissions (Duncan et al., 2010; Martin et al., 2004). Similarly, $PM_{2.5}$ $NO_3^-$ sensitivity to $NO_x$ versus $NH_3$ can be diagnosed using $NH_3$ to $NO_2$ ratios measured from satellites (Dang et al., 2023, 2024)."*

4. Line 87-95: This paragraph would be a good place to highlight the purpose and novelty of your work in a brief statement.

   Revised as follows:
   *"Here we analyze recent 2015−2023 trends in air quality in Korea by exploiting surface, airborne, and satellite observations to provide insights for the effectiveness of past regulation policies and future management."*

5. Table 1: What about the temporal resolution or overpass time of the satellites? The LEO orbit satellites will be measuring at a specific time of day over South Korea, so they could be catching the daily min or max values. How does this compare to the times other measurements are available for? A discussion of how this might affect the differences in the results for different datasets is missing. This could be included in the relevant results sections or section 2.

   We included detailed descriptions on satellite overpass times and how the data are sampled in the text (Lines 125-131) and figures.

*"OMI and TROPOMI make afternoon overpasses at 13:30 local time (LT). We make use of morning overpasses for MOPITT (10:30 LT) and IASI (9:30 LT). We use hourly daytime observations from GEMS (7:45−16:45 LT), GOCI (9:30−16:30 LT), and GOCI-II (8:15−17:15 LT). For annual trend analyses we use GEMS observations made between 12−14 LT for consistency with the overpass time of OMI and TROPOMI measuring the same gases. We find no significant differences in observed trends when using surface observations sampled at satellite overpass times and therefore use all hours of the day."*

6. Line 149: Expand on how the topography affects the CO along the east coast
   Revised as follows:
   *"Low VCDs over mountainous areas are due to surface elevation reducing the background column. This effect of surface elevation on VCDs is less apparent for shorter-lived species with weaker background contributions."*

7. Line 158: The 2019 spike seems quite small – is it greater than the uncertainty of the data?
   The retrieval uncertainties of CO from MOPITT and TROPOMI are $0.6\text{-}1\times10^{17}$ molec. cm$^{-2}$, and this spike is $0.9\times10^{17}$ molec. cm$^{-2}$, which appears to be insignificant as pointed out and removed this sentence.

8. Line 165: Specify what the continuing motivation for SO$_2$ emission controls is
   Revised as follows:
   *"However, there is continuing motivation for emission controls because SO$_2$ is a precursor to PM$_{2.5}$ sulfate (SO$_4^{2-}$)."*

9. Lines 174-177: Link the other studies' results back to your findings, e.g., are they consistent, what are the implications of the different sources of SO$_2$
   We find consistent decreases in 2016-2022 surface SO$_2$ from our analysis, therefore revised as follows:
   *"J. Park et al. (2024) found that recent trends (2016–2022) in national mean surface SO$_2$ were driven by reductions in both domestic (25%) and Chinese (16%) emissions, explaining the 41% decrease shown in Figure 3d."*

10. Line 218: Clarify why the transportation contribution may be a severe underestimate
    VOC source apportionment studies over Korea listed in the reference report a 20-30% contribution from vehicle emissions. S. Song et al. (2019) explain the underestimate in the inventory through discrepancies in the source profile used which depends on car type, driving conditions, emission control technology, etc. We clarified the degree of underestimate as follows:
    *"More than half of anthropogenic VOC (AVOC) emissions according to CAPSS are from solvent use while transportation is responsible for less than 10%, although the latter may be underestimated by a factor of 2–3 according to source apportionment studies (S. Song et al., 2019; Y. Kim and G. Lee, 2018; Kwon et al., 2021)."*

11. Line 250: "values in Korea are higher everywhere" is inconsistent with previous statements. Below 0.03 can be greater than 0.02.
    Revised as follows:
    *"GEMS R$_{GF}$ values in Korea are higher than 0.01 everywhere, indicating a more important role for AVOC emissions than in the US where these emissions have been strongly regulated for decades (Parrish et al., 2009; Warneke et al., 2012)."*

**12.** Line 254: Although there is no significant trend in surface BTEX, can you comment on the higher values over 2019-2021? Is this within the data uncertainty or a significant signal?

For annual trend analyses we used observations with complete 2015-2023 records so that each year contains data from identical monitoring sites. We first clarified this in Section 2 (Lines 109-110) and in figure captions:

*"For annual trend analyses we use observations from AirKorea sites that have continuous records from 2015 to 2023."*

In the current Figure 5g we averaged data across all BTEX AirKorea sites, which resulted in large interannual variability and no significant trend. We thank the reviewer for pointing this out and corrected Figure 5g, which now shows a decreasing trend in BTEX. We revised the manuscript as follows:

*"Figure 5g shows no significant trends in AVOC emissions and satellite observations of CHOCHO and HCHO, although surface BTEX decreased at $-5.0 \pm 3.9\%\ a^{-1}$ during 2015−2023."*

**13.** Line 269: Can you comment on why the satellite data do not show the late afternoon rise?

Differences may arise from the flight track dependency of aircraft data (Kwon et al., 2021). We revised as follows:

*"The aircraft data show a late afternoon rise in HCHO for which we have no explanation and might reflect sparse sampling (Kwon et al., 2021)."*

**Reference:** Kwon et al., Elementa 2021 (https://doi.org/10.1525/elementa.2021.00109)

**14.** Line 279: This seems to be the only result for 2005-2014 in the paper. Is it relevant to the rest of the work? If not, I would suggest removing it.

We agree and revised as follows:

*"We find that May–June 90$^{th}$ percentile MDA8 $O_3$ calculated for individual AirKorea sites and then averaged across all sites shows an increase of $0.8 \pm 0.9\ ppbv\ a^{-1}$ during 2015–2023 (Figure 7b)."*

**15-16.** Line 279: Can you comment on the $O_3$ change between 2019 and 2020? Lines 301-305: Link the US data back to your results, otherwise they just read as additional, slightly random, facts.

Thank you for pointing this out. We removed the US trends and revised the paragraph focusing on background $O_3$ and its affect during 2019-2020. Previous studies confirmed that changes in domestic $NO_x$ or VOCs emissions during the COVID-19 pandemic did not have significant impacts on pollutant levels (S.-W. Kim et al., 2023; Koo et al., 2020). Instead, emission reductions in China during the lockdown reduced long-range transport and contributed to the observed changes in Korea. We revised the manuscript as follows (Lines 320-331):

*"Reports of $O_3$ increases in Korea based on data from the AirKorea sites may be biased by the AirKorea sites being concentrated in the SMA, which has been mostly VOC-limited, but this is now changing as $NO_x$ emissions decrease. Our analysis suggests that $O_3$ pollution in Korea is now poised to decrease everywhere in response to continued $NO_x$ emission controls.*

*An additional challenge for Korea to meet its air quality standard is the high background originating from East Asia, estimated to be 55 ppbv (Colombi et al., 2023). During the COVID-19 lockdown in 2020 precursor emissions significantly dropped in China but not in Korea (Koo et al., 2020), which led to reduced long-range transport of $O_3$ and hence lower background levels over Korea (S. W. Kim et al., 2023). This could explain the large decrease in $O_3$ found between 2019 and 2020, especially in $NO_x$-limited areas which are more sensitive to background contributions than local emissions."*

**Reference:** Koo et al., Sci Rep 2020 (https://doi.org/10.1038/s41598-020-80429-4)

**17.** Figures 2-9: It would be useful to see some measure of uncertainty or error on the line graphs, or a statement on the associated uncertainty in the main text.

Retrieval uncertainties of satellite observations vary depending on how the uncertainties are defined and therefore we only commented on this for the $SO_2$ VCDs which have largest uncertainties (Lines 189-190). For all other data we show the error standard deviations of the regression slopes (trends).

**18.** Additional detail on data analysis could be added to the supplement or processing scripts shared in a code availability section

We added details on how the surface and satellite data where sampled, averaged, and analyzed in the Supplement.

**Technical corrections:**

We have revised the manuscript as suggested.

**1.** Line 40: Clarify 'Subsequent atmospheric chemistry (of these trace gases?) produces'
**2.** Line 68-69: I would change 'would respond nonlinearly' to 'responds nonlinearly', as this a general statement
**3.** Line 79: clarify you are listing the relevant LEO instruments
**4.** Line 116: "We do not use them here." Be explicit that the $O_3$ measurements are what is not used. This sentence and the previous one could be combined for clarity: "(…) are inconsistent over Korea (Gaudel et al., 2018), therefore we do not use them here."
**5.** Line 135: Explain 'SMA' acronym in the main text
**6.** Line 144: 'plays an important role in driving ozone formation'
**7.** Line 169-170: I suggest rephrasing to "(…), consistent with OMI $SO_2$ hotspots previously identified for 2011-2016 (Chong et al., 2020)." for easier reading.
**8.** Line 182: replace 'accounting' with 'which account' for clarity
**9.** Line 184: add 'the potential' to match 'motivated by': "but also the potential to reduce PM2.5"
**10.** Line 185: replace 'diesel engines in 2016' with 'diesel engines since 2016'
**11.** Line 198: "CAPSS shows an increase"
**12.** Line 202: "additional information on the diurnal variation of $NO_2$"
**13.** Line 320: "and is at its minimum in summer"
**14.** Line 327: "$PM_{2.5}$ observations in Seoul show" (not shows)
**15.** Line 386: "(…) component not found to show"
**16.** Line 393: "for in terms of decreasing $O_3$"